# Conplastic FVB/N-mt129S6/SvEvTac mice: A new tool for cancer research

Artiom Gruzdev[1], Wendy N. Jefferson[1], Thomas B. Hagler[1], Gregory J. Scott[1], Manas K. Ray[1], Ginger W. Muse[2], Rani S. Sellers[3], Carmen J. Williams[1]*

1 Reproductive & Developmental Biology Laboratory, National Institute of Environmental Health Sciences, National Institutes of Health, Durham, North Carolina, United States of America, 2 Epigenetics & RNA Biology Laboratory, National Institute of Environmental Health Sciences, National Institutes of Health, Research Triangle Park, North Carolina, United States of America, 3 Department of Pathology and Laboratory Medicine, School of Medicine, University of North Carolina at Chapel Hill, Chapel Hill, North Carolina, United States of America

* williamsc5@niehs.nih.gov

## Abstract

FVB/N mice, which are commonly used for cancer studies, have accelerated onset of endometrial cancer following developmental estrogenic chemical exposure. These mice also have a polymorphism in the mitochondrial gene, *mt-Atp8*, leading to increased production of reactive oxygen species. We hypothesized that this polymorphism contributes to the enhanced endometrial cancer phenotype in FVB/N mice. To test this idea, we generated conplastic FVB/N-mt129S6/SvEvTac mice (FVB/N nuclear genome; 129S6/SvEvTac mitochondria: FVB/N-mt129). The impact of 129S6 versus FVB/N mitochondrial genomes on endometrial cancer development following neonatal exposure to the xenoestrogen, diethylstilbestrol, was tested by comparing the cancer phenotypes of FVB/N mice to FVB/N-mt129 mice. There was no difference in cancer incidence regardless of mitochondria source, but cancer grade was higher in the conplastic strain. Additionally, while the FVB/N genetic background is considered non-permissive for generation of pluripotent mouse embryonic stem cells, blastocysts from the conplastic background readily generated mouse embryonic stem cell clones that supported gene editing in culture and subsequently generated germline competent chimeric founder mice. FVB/N-mt129 mice are a potentially powerful resource for generating germline competent embryonic stem cells with an FVB/N nuclear genome and for studying cancer phenotypes.

## Introduction

Endometrial cancer is the sixth most common cancer in women globally, and the most common cancer of the female reproductive tract; its incidence continues to increase [1,2]. Endocrine disruptors and estrogenic compounds are important risk factors in the development of endometrial cancer, with early exposure potentially

**Data availability statement:** All relevant data are within the manuscript and its Supporting information files.

**Funding:** This research was supported by the Intramural Research Program of the National Institutes of Health (NIH), National Institute of Environmental Health Sciences, 1ZIAES102405 (CJW) and 1ZICES102425 (Gene Editing and Mouse Model Core). The authors would like to thank the Molecular Genomics Core (1ZICES102546) and the Comparative Medicine Branch Animal Research Support Core (1ZIGES102585) for their support. The contributions of the NIH author(s) are considered Works of the United States Government. The findings and conclusions presented in this paper are those of the author(s) and do not necessarily reflect the views of the NIH or the U.S. Department of Health and Human Services. The funders had no role in study design, data collection and analysis, decision to publish, or preparation of the manuscript.

**Competing interests:** The authors have declared that no competing interests exist.

exacerbating risk in younger women and girls [3,4]. Hormone signaling involves multiple organ systems and the initiation of reproductive cancers often occurs while the reproductive tract is actively developing, necessitating the use of *in vivo* cancer models. While some animal models of cancer can be highly variable, the neonatal mouse model of endometrial cancer induction using the estrogenic chemical, diethylstilbestrol (DES), is a consistent and predictable model of endometrial carcinogenesis. DES-induced endometrial cancer does not correlate with any of the published human endometrial cancer molecular subgroups [5]; it is an endometrioid adenocarcinoma with no known mutational signature. Molecular characterization of this cancer using single cell RNA-seq showed widespread activation of Wnt/β-catenin signaling in epithelial cells and abnormal PI3K/AKT activation in cancer cells [6].

There is notable mouse strain variation in the onset, penetrance, and severity of DES-associated endometrial cancers. For example, CD-1 outbred mice administered DES as neonates do not develop cancer until they are older than 8 months of age, and the incidence is only 47% at 12 months of age [7]. Various C57BL/6 sub-strains are one of the most commonly used mouse lines in biomedical research, and they develop basal cell metaplasia, squamous metaplasia, and endometrial hyperplasia following neonatal exposure to DES, but lesions do not progress to endometrial cancer [8]. FVB/N mice have favorable reproductive characteristics and are commonly used for mouse cancer models [9,10]. We find that FVB/N mice develop endometrial carcinoma following the same histological progression as CD-1 mice, but in an accelerated time frame, with an endometrial cancer incidence of 46% at 8 months, 73% at 12 months, and 93% at 18 months of age [11,12]. These characteristics make FVB/N mice a useful model for studying endometrial carcinogenesis, and unlike CD-1 outbred mice, the inbred genome of FVB/N mice minimizes variability.

The importance of reactive oxygen species (ROS) during the development and progression of cancer is well documented [13–15]. Estrogen increases the generation of ROS and has been implicated in the development of breast and endometrial cancer in women [16,17]. Interestingly, FVB/N mice harbor a variant in the mitochondrial gene *mt-Atp8* (mt-ATP8*D5Y; nucleotide position 7778 G-to-T), leading to altered mitochondrial structure, an elevated ATP/ADP ratio with normal ATP levels, and increased mitochondrial ROS production [18]. Similarly in pancreatic islets, the *mt-Atp8*D5Y* polymorphism does not affect overall cellular ATP levels but leads to an increase of mitochondrial ROS indicating that the variant's impact does not result in severe mitochondrial dysfunction, just altered mitochondrial function [19]. This altered mitochondrial function leading to increased ROS, which by itself has been shown to impact cancer development, led to our hypothesis that the increased susceptibility of FVB/N mice to DES-induced endometrial carcinoma is at least partially attributed to the *mt-Atp8*D5Y* mitochondrial variant.

To test this idea, we generated a conplastic strain of FVB/N mice that carried 129S6/SvEvTac mitochondria. The 129S6 genetic background was chosen because it has genetic similarity to the 129/Ola genetic background from which the E14 mouse embryonic stem (ES) cell line was generated; E14 was one of the first isogenic ES cell lines that had consistently high potential for germline contribution [20]. 129/Ola

isogenic mice no longer exist, so 129S1 (Jackson Laboratory), 129S2 (Charles River Laboratories), and 129S6 (Taconic) mice were considered. All three were relatively similar to each other and comparably divergent from the 129/Ola background. This new mouse model was compared to standard inbred FVB/N mice to determine relative timing of development of endometrial cancer following neonatal DES exposure. Finally, we tested the efficiency of germline transmission of the ES cells derived from the conplastic mice both before and after genetic modification to determine their utility in future mechanistic studies.

## Materials and methods

### Biochemicals and reagents

All biochemicals and other reagents were obtained from Sigma-Aldrich (St. Louis, MO) unless otherwise specified.

### Animals and husbandry

All animal studies were performed under NIEHS Animal Care and Use Committee-approved protocols (RDBL08–22 and RDBL07–38). Studies were carried out in the NIEHS animal facility in compliance with the US National Research Council's Guide for the Care and Use of Laboratory Animals. FVB/NJ mice (strain #001800; RRID:IMSR_JAX:001800), C57BL/6J mice (strain #000664; RRID:IMSR_JAX:000664) and albino C57BL/6J female mice [B6(Cg)-$Tyr^{c-2J}$/J; strain #000058; RRID:IMSR_JAX:000058] were purchased from Jackson Labs. 129S6/SvEvTac mice (model no. 129SVE-F; MGI:2161090) were purchased from Taconic and Swiss Webster mice (strain Crl:CFW; MGI:5911387) were purchased from Charles River. Mice were housed in Tecniplast individually ventilated cages in a specific pathogen-free barrier facility. Mice were provided 5K20/5KON diet (LabDiet) and water ad libitum. Rooms were maintained at 65–75°F with 40–60% humidity on a 12-hour light and dark cycle. In accordance with approved protocols indicated above, all mice were euthanized at their predetermined end point. There was no expectation of premature death due to experimental conditions; however, all mice were monitored for general health and behavior twice per week during routine animal husbandry to ensure their health. All animal welfare considerations were taken to minimize suffering and distress.

### Generation of FVB/N-mt129 conplastic mice

FVB/NJ males were crossed with 129S6/SvEvTac females to generate 129FVBF1 offspring. Female 129FVBF1 were then crossed to FVB/NJ males to generate the N2 generation of mice with 129S6/SvEvTac mitochondria. Females were then backcrossed for 17 generations (N17) into commercial FVB/NJ males to create an FVB/NJ nuclear genome mouse with fully functional 129S6/SvEvTac mitochondria: conplastic FVB/N-mt129 mice (FVB/N-mt129S6/SvEvTac; referred to henceforth as FVB/N-mt129). At generation N9, when the nuclear genome was theoretically expected to be 99.8% FVB/N, the FVB/N-mt129 conplastic line was analyzed by whole genome miniMUGA SNP scan (NeoGen, Lansing, MI, USA). At generation N10 (theoretically 99.9% FVB/N nuclear genome), ES cells were derived from naturally mated FVB/N-mt129 mice (see detailed methods below). The conplastic FVB/N-mt129 mice will be available to the research community upon acceptance of the manuscript.

### Derivation of FVB/N-mt129 ES cells

Embryonic day 3.5 (E3.5) embryos were collected from FVB/N-mt129 females mated to FVB/NJ males. Embryos were collected and washed in DMEM (Cat no: 11965−092; ThermoFisher Scientific, Waltham, MA, USA) with 10% FBS (ES cell certified) and placed in individual wells of 96 well plates with irradiated DR4 mouse embryo fibroblast feeder cells (A34966, ThermoFisher Scientific) in ES cell culture media with the following composition: 80% DMEM, 15% FBS, 2 mM L-glutamate, 100 mM 2-mercaptoethanol, 0.1 mM MEM Non-Essential Amino Acids Solution (Cat no: 11140050; ThermoFisher Scientific), 1 mM sodium pyruvate, 1,000 U/ mL leukemia inhibitory factor (LIF), penicillin (100 U/mL),

streptomycin (100 µg/mL), 1 µM PD0325901 (MEK inhibitor; ReproCell USA, Beltsville, MD, USA), and 3 µM CHIR99021 (GSK-3 inhibitor; ReproCell). Plates were incubated at 37°C in 5% $CO_2$ and 5% $O_2$. After 1 week, embryos were dissociated and moved to a new 96 well plate with irradiated mouse embryo fibroblast feeder cells. From the second 96 well plate, individual wells were monitored for confluency and passaged as needed on gelatin coated tissue culture dishes without additional irradiated feeder cells: 96 well to 24 well to 6 well to final 60 mm dishes. Cultures were frozen down at the 60 mm dish stage, with a small aliquot of cells returned to culture in 24 well dishes for genetic analysis. Individual FVB/N-mt129 clones were genotyped for *Sry* to identify XY karyotype clones. XY clones were confirmed to be *Tyr^C* homozygous and then submitted for whole genome miniMUGA SNP scan.

## Pluripotency assessment of FVB/N-mt129 ES clones

Five independent FVB/N-mt129 ES clones were microinjected into E3.5 C57BL/6J blastocysts isolated from naturally mated C57BL/6J males and females as follows. ES cells were dissociated and suspended in ES-culture media as above and placed on ice. Microinjections were done in a 50 mm glass bottom dish (MatTek Corp, cat #P50G-0–30-F) containing DMEM with 20 mM HEPES and 10% FBS, using lab-pulled borosilicate glass holding pipettes. ES cells were picked up individually using TransferTip (ES) injection needles (Eppendorf, Hamburg, Germany). Approximately 15 ES cells were injected into the blastocoel of each embryo. After injection, 15–20 embryos were non-surgically transferred transcervically using the Non-Surgical Embryo Transfer Device (Paratechs Corp, Lexington KY, USA) into E2.5 pseudopregnant Swiss Webster female mice previously mated to vasectomized Swiss Webster male mice. Male chimeric founders were then bred to albino C57BL/6J females to determine germline transmission of the FVB/N-mt129 ES cell genome. Albino offspring were genotyped for both the FVB/N-derived *Tyr^C* (TYR*C103S) mutation and the C57BL/6J-derived *Tyrc-2J* (TYR*R77L) mutation and then submitted for whole genome miniMUGA SNP scan.

## Gene editing in FVB/N-mt129 ES cells

To confirm pluripotency after *in vitro* gene editing, the *Tyr^C* mutation (TYR*C103S) in the FVB/N genome was edited to wild type tyrosinase-expressing sequence using CRISPR/Cas9 gene editing. Cas9 was targeted to AACTGCG-GAAACTcTAAGTT NGG[PAM] of the *Tyr^C* allele; the C103S-causing mutation is shown in lower case. To disrupt the CAS9 target sequence in the homology directed repaired allele and to facilitate screening, synonymous mutations were introduced in the adjacent codons, which added unique Mfe I restriction sites (Fig 1). In parallel, two different species of repair templates were used to generate the same *Tyr^C*-to-WT allele: A plasmid with 654 bp of total homology, and a ssODN with 100 bp of total homology. The plasmid was generated via conventional cloning and site directed mutagenesis with the following homology arms (HA): 5′-HA (349 bp – chr7:87,142,263−87,142,607 mm39) and 3′-HA (309 bp – chr7:87,141,935−87,142,243). The ssODN with 50 bp homology arms (5′-chr7:87,142,263−87,142,312 and 3′-chr7:87,142,194−87,142,243) was obtained from Integrated DNA Technologies Inc (Coralville, IA, USA).

Three independent, germline competent FVB/N-mt129 ES clones were transfected with a 6:1 molar ratio of donor plasmid or ssODN and Cas9-Puro/sgRNA delivery plasmid [pSpCas9(BB)-2A-Puro (PX459) V2.0], a gift from Feng Zhang (Addgene plasmid #62988) [21]. After transfection, the cells were exposed to 48 h of puromycin selection (0.9 µg/mL) followed by standard clonal expansion/screening. Clones were screened by PCR with primers external to both homology arms (Fwd: 5′-CATGTGCTTTGCAGAAGATAAAAGC-3′, Rev: 5′-TCCCCAGTTAGTTCTCGAATTTCTT-3′). Homozygous clones were identified by Mfe I restriction digest and confirmed by amplicon sequencing. For the chimeric founder generation, six independent homozygous ES clones (one per starting FVB/N-mt129 clone per repair template type) were microinjected in two separate microinjection sessions into E3.5 albino C57BL/6J blastocysts isolated from naturally mated females and males. Micro-injected blastocysts were then non-surgically transferred to pseudo-pregnant (E2.5) recipient Swiss Webster mice as described above. Of the six injected clones, all six yielded coat color contribution chimeras, with three different clones readily resulting in male germline transmitting chimeras. Once three different gene

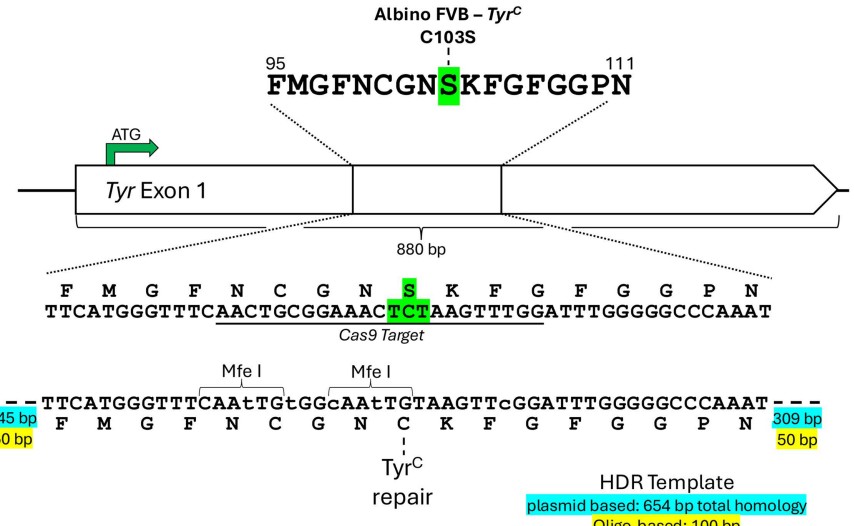

**Fig 1. Gene editing strategy to repair the albinism *Tyr^C* mutation found in FVB/N mice.** Targeting strategy for correction of the *Tyr^C* albino point mutation (green) utilizing a sgRNA/Cas9/Puromycin resistance delivery plasmid co-transfected with either a plasmid donor (blue) or ssDNA oligo donor (yellow). Silent mutations were introduced into neighboring codons for screening purposes.

edited FVB/N-mt129 ES clones were confirmed to be germline competent, all additional chimeric founder breeding was halted. The first germline male chimera founders were then mated to the parental FVB/N-mt129 females to generate the FVB/N-mt129 conplastic mouse line, which has agouti coat color in otherwise FVB/N nuclear genome with 129S6 mitochondria. The agouti FVB/N-mt129 mouse line was then submitted for whole genome miniMUGA SNP scan.

## Mitochondria sequencing

Total DNA (nuclear and mitochondrial) was isolated from tail biopsies using proteinase K digestion. The mitochondrial genomes from three independent mice from each strain were sequenced using staggered PCR amplicon Sanger sequencing as described previously [22].

## Whole genome SNP scan

For whole genome SNP scans, 3 µg of genomic DNA from ES clones or mouse tail biopsies were submitted to Transnetyx (Cordova, TN, USA) and the final miniMUGA platform analysis was performed by NeoGen (Lansing, MI, USA). As of 2024, the miniMUGA platform assessed 9848 autosomal SNPs, 763 SNPs on the sex chromosomes (including the pseudoautosomal regions), and 88 SNPs in the mitochondrial genome. C57BL/6J genetic background variants were included in the miniMUGA analysis to confirm the lack of contamination from the most common other genetic background housed in the same animal breeding room.

## Induction of endometrial cancer using diethylstilbestrol

At 2 months of age, FVB/NJ and FVB/N-mt129 females were bred to 2-month-old wild type FVB/NJ males to generate female pups. Pups were sexed shortly after birth and 8–10 female pups were combined and placed with dams for subsequent treatment; males were humanely euthanized. To induce endometrial cancer, female pups were administered 1 mg/kg/day (2 µg/pup/day) diethylstilbestrol (Sigma cat# D4628) in 0.02 mL corn oil (Spectrum Chemical Manufacturing Company, cat# C0136) subcutaneously for 5 days from postnatal days 1–5 as described previously; no controls

were included as they do not develop uterine cancer (7). An *a priori* power calculation of the number of mice needed to observe an absence of uterine cancer in the FVB/N-mt129 line compared to the FVB/NJ line with 80% confidence was 22 mice. Mice were weaned at postnatal day 22 and housed in groups of 5 mice per cage. Mice were provided NIH31 diet (Zeigler Feed Manufacturing) and water ad libitum with daily light cycles of 12 hours of light and 12 hours of dark. To capture differences in uterine cancer incidence (decreased or increased), we selected 9 months of age as our timepoint. This timepoint should result in >50% incidence of uterine cancer in the wild type FVB/N based on previous studies using this model, which showed uterine cancer incidence of 46–49% at 8 months and 72–73% at 12 months [11,12,23]. Mice were humanely euthanized at 9 months of age. Uteri were collected, fixed in 10% neutral buffered formalin for 48 h, and subsequently placed in 70% ethanol (FVB/N mice, N = 24; FVB/N-mt129 mice, N = 42); all weaned mice were included in this study. Of note, one litter of female pups (n = 8) in the FVB/N group was lost prior to weaning due to a flooded cage and was not included in the study. Samples were routinely processed to paraffin, sectioned at 6 µm, and stained with hematoxylin and eosin by the NIEHS Pathology Support Group. Two sections 36 µm apart were microscopically evaluated.

## Histological analysis

A masked histological analysis was performed by a board-certified veterinary pathologist (RSS). Two coronal uterine sections per mouse were assessed based on parameters identified previously [24]. Both sections were scored and were typically not different, but in case of differences the more severe score was recorded. Each histological section included cervix, uterine body, one full uterine horn, and a small portion of the contralateral horn. Squamous metaplasia was graded on a 0–5 scale based on extent (0 – not present; 1 – minimally present; 2 – mildly present; 3 – moderately present; 4 – markedly present; 5 – severely present). Basal cell metaplasia was graded on a 0–5 scale in severity, according to the following parameters. Grade 1 correlated with basal cell metaplasia identified in 0.5 100x fields. Grade 2 correlated with basal cell metaplasia in most/all glands in 1 to 1.5 100x fields. Grade 3 correlated with basal cell metaplasia in most/all glands in 2–3 100x fields. Grade 4 correlated with basal cell metaplasia in most/all glands in 4–5 100x fields. Grade 5 correlated with basal cell metaplasia in all glands in 6–7 100x fields. Tumors were graded based on the extent of tumor distribution ranging from no tumor to extensive tumor throughout the uterine body/cervix and horn(s). Using this grading system, tumor grade 1 reflected the presence of a small focus of carcinoma that was localized only to the uterine body. Tumor grade 2 reflected either a large focus of tumor in the uterine body ± cervix or small tumor foci in the body of the uterus and at the most proximal end of one uterine horn (close to uterine body). Tumor grade 3 reflected a large amount of tumor in the uterine body/cervix plus tumor in the uterine horn extending closer to the middle of the horn or was present at the proximal end of both horns. Tumor grades 4 and 5 were similar, with uterine body/cervix tumor plus horn tumor. Grade 4 had tumor present less extensively through the entirety of one horn and into the residual horn; grade 5 was present and extensive in the uterine horn and residual horn, extending to the most distal end of the horn. Examples of phenotypic features are provided (Fig 2). A full description of the histological features of DES-induced endometrial cancer are published [24].

## Statistical analysis

To determine the sample size needed for the cancer study, we performed an a priori calculation to detect a significant reduction in uterine cancer incidence between the two lines. Using an 80% confidence level and assuming at 9 months of age a cancer incidence of 50–60% in the FVB line and 0–20% in the FVB-129mt line (similar to that in CD-1 mice) [7,25], we needed 11 mice per group (50% versus 0%) or 22 mice per group (60% versus 20%). Comparisons of cancer incidence were made using a Fisher's exact test; only presence or absence was used for this calculation. Comparisons of phenotypic grade were made using the Chi square test for trend. Analyses were done using GraphPad Prism (version 9.5.1).

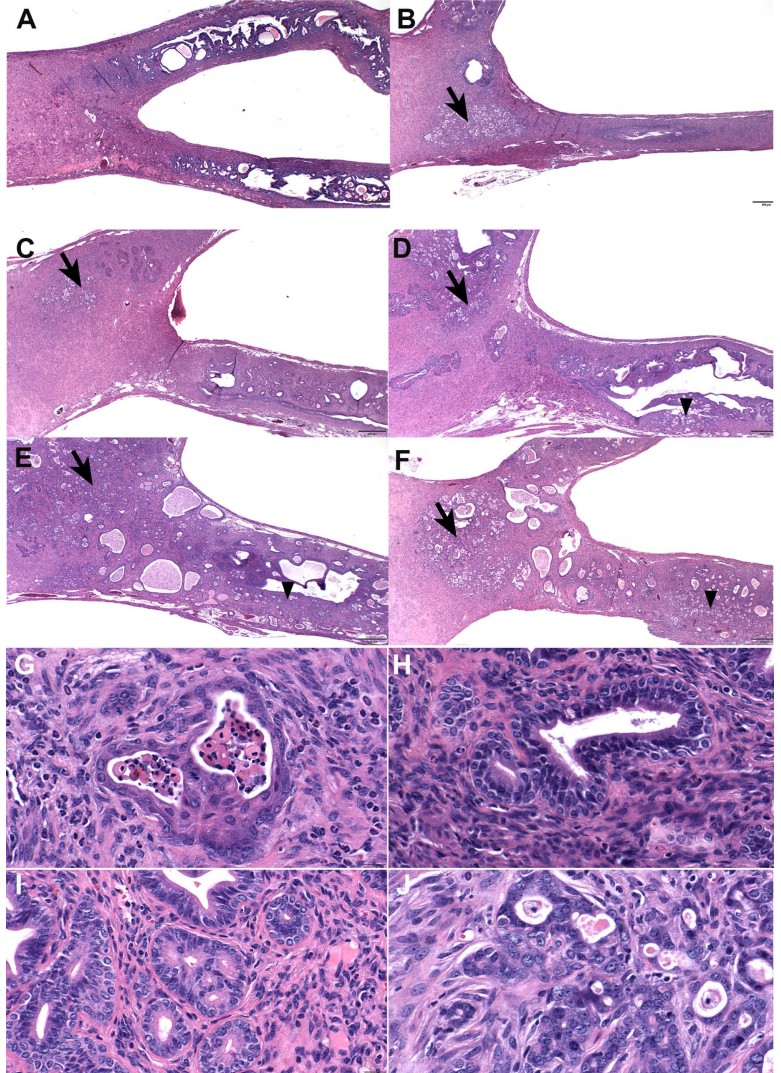

**Fig 2. Microscopic grading of uterine tumor burden.** Tumors were graded from 0-5. (A) Grade 0 – only cystic endometrial hyperplasia evident. (B) Grade 1: small focus of carcinoma that was localized only to the uterine body (arrow). (C) Grade 2: large focus of tumor in the uterine body±cervix or small tumor foci in the body of the uterus and at the most proximal end of one uterine horn (close to uterine body) (arrow). (D) Grade 3: large amount of tumor in the uterine body/cervix (arrow) plus tumor in the uterine horn extending closer to the middle of the horn or was present at the proximal end of both horns (arrowhead). Grades 4 (E) and 5 (F) were similar, with uterine body/cervix tumor plus horn tumor. Grade 4 had tumor present less extensively through the entirety of one horn and into the residual horn; grade 5 was present and extensive in the uterine horn and residual horn, extending to the most distal end of the horn. (G) Focus of glandular squamous metaplasia in uterine horn with desquamated cells and neutrophils in the lumen. (H) Focus of glandular basal cell metaplasia; note the cuboidal layer of epithelial cells underlying the tall cuboidal to columnar epithelium. (I) Glandular epithelium, atypical hyperplasia. (J) Neoplastic glandular epithelial cells infiltrating through the uterine stroma. H&E. A-F=20x magnification; G-J=400x magnification.

## Results and discussion

Taking advantage of mitochondrial inheritance only from the maternal line, we crossed FVB/NJ males with 129S6/SvEvTac females to generate 129FVBF1 offspring that had 100% 129S6/SvEvTac mitochondria. Female offspring were then crossed to FVB/NJ males to produce subsequent generations of mice that maintained only 129S6/SvEvTac mitochondria. The favorable reproductive characteristics of FVB/N mice were maintained in the FVB/N-mt129 conplastic mouse line, which had 11.5±3.6 pups/litter (N=53 litters).

Generation of the FVB/N-mt129S6/SvEvTac conplastic mouse line (at N10) was confirmed by medium resolution whole genome single-nucleotide polymorphism (SNP) scan. The raw SNP results from ten representative conplastic FVB/N-mt129 mice are listed in Supplemental S1 Table. All ten mice matched the FVB/N genotype in 99.1% (8865/8830) of the locations tested. In 0.7% (62/8830) of the locations, all mice that had a reported genotype matched FVB/NJ, but some mice failed to yield a genotype call for the specific variant. In only 0.2% (18/8830) of the locations, some mice had reported variants that were not consistent with FVB/NJ genotype. These findings indicate that the nuclear genomes of the FVB/N and FVB/N-mt129 mice were essentially identical.

The mitochondrial genome as sequenced in the FVB/N mice in our colony was 100% as expected for FVB/N mice [GenBank EF108338.1 [26] and EMBL-EBI/Wellcome Sanger Institute GenBank OW971719.1]. The 129S6/SvEvTac mitochondrial genome sequence has not been deposited in any open access online databases but based on the phylogeny of the various 129 Steel mouse lines, we expected a mitochondrial genome relatively similar to the published 129S1 mitochondrial genome (GenBank EF108330). Relative to the 129S1 mitochondrial genome, the 129S6 parental mice used in this study had two heterozygous mitochondria variants identified, one of which was synonymous (Table 1). The non-synonymous heterozygous variant in *mt-Atp6* that results in mt-ATP6*N183S has unknown consequences for mitochondrial function, but it is likely benign based on the INPS-MD (Impact of Non-synonymous mutations on Protein Stability – Multi Dimension) [27] prediction that it has no impact on stability. Furthermore, the mt-ATP6*N183S variant is not in a highly conserved region of mt-ATP6 and is found in other mouse species such as *Mus terricolor* (NCBI BioProject: PRJNA927338).

In the genetically modified mouse line community, the FVB/N genetic background is known to be non-permissive to the generation of germline competent ES cells [28]. A possible explanation for this finding is that the increased ROS production by FVB/N mitochondria predisposes the ES cells to differentiate, limiting their capacity for germline transfer [29]. For this reason, we tested if FVB/N-mt129 derived ES cells could be genetically modified and support the generation of live mice. Similar to most 129 Steel-derived and 129 parental-derived mouse lines, the FVB/N-mt129 conplastic mouse line readily produced ES cells using blastocyst plating methodology. Conplastic ES cells were generated from blastocysts obtained from N11 FVB/N-mt129 mice in 2 independent sets, resulting in 29% and 35% successful establishment from a single blastocyst (14/48 and 21/60 blastocysts respectively). By medium resolution whole genome SNP scan, the N11-derived conplastic ES cell clones were virtually identical to the N10 conplastic mouse line, with minor differences between samples attributed to technical issues. Five XY FVB/N-mt129 ES cell clones were tested for pluripotency/germline potential, three of which generated germline transmitting male chimeras and were used for subsequent genome editing studies. A potential limitation of our study is that we did not attempt in parallel to generate germline capable isogenic FVB/N mouse ES cells. Given the anecdotal evidence that FVB/N is a non-permissive strain, this would have been cost prohibitive with potential ethical issues in regard to research animal use.

**Table 1. Mitochondrial genome variants in the conplastic FVB/N-mt129 mouse line.**

| mtDNA position (bp) | Gene | Parental FVB/N strain | Conplastic FVB/N-mt129 strain | 129S1 strain (Reference) | Mitochondrial Functional Impact |
|---|---|---|---|---|---|
| 7778 | *mt-Atp8* | T (mt-ATP8*D5Y) | G | G | D5Y linked to ROS generation and mitochondrial dysfunction (18) |
| 8474 | *mt-Atp6* | A | A/G (heterozygous) (mt-ATP6*N183S) | A | Unknown; INPS-MD [27] predicts ATP6*N183S to have no impact on protein stability. |
| 13583 | *mt-Nd6* | A | A/G (heterozygous) (synonymous) | A | No impact |
| 15124 | *mt-Cytb* | A | G (mt-CYTB*I327V) | G | Unknown; INPS-MD [27] predicts CYTB*I327V to have no impact on protein stability. |

As a proof of concept, we subjected FVB/N-mt129 ES cell clones to CRISPR/Cas9 gene editing of the *Tyr^C* mutation, resulting in the easy-to-screen phenotype of the presence or absence of albinism. Using either a single-stranded oligodeoxynucleotide (ssODN) or a plasmid-based repair template, the FVB/N-mt129 ES cell clones were readily transfectable and capable of clonal expansion following puromycin selection. Gene editing efficiency of FVB/N-mt129 ES cell clones was similar to that observed in other standard ES cell lines used by our group. These germline competent FVB/N-mt129 ES cells will allow for the production of genetically modified mice containing a 100% pure FVB/N genome but without the need for time-consuming backcrossing. Thus, we have generated a powerful conplastic tool that can be used to accelerate research using FVB/N mouse models.

We next asked if the conplastic FVB/N-mt129 mice, which lack the mitochondrial variants thought to drive excessive ROS production in FVB/N mice, had differences in their susceptibility to DES-induced endometrial cancer. FVB/N or FVB/N-mt129 female pups were exposed to DES daily on postnatal days 1–5 and aged to 9 months of age. A histopathological assessment of the uteri from these 9-month-old females was then performed, scoring the extent of basal cell metaplasia, squamous cell metaplasia, and adenocarcinoma. The incidence and severity/extent of basal cell metaplasia and squamous metaplasia were not different between groups (Fig 3A, B and Table 2). Adenocarcinoma was present in 98% (41/42) of the FVB/N-mt129 group and 83% (20/24) of the FVB/N group; this difference was not significant (Fisher's exact test, $p = 0.0547$). However, the adenocarcinoma grade was higher in the FVB/N-mt129 group relative to the FVB/N group (Fig 3C and Table 2; Chi square test for trend, $p = 0.0023$). Overall, these findings indicate that there is a slightly worse cancer phenotype in the conplastic FVB/N-mt129 group rather than in the group with mitochondrial dysfunction. We conclude that the FVB/N mitochondrial variant is not a primary driver of endometrial cancer susceptibility in this strain, but instead the susceptibility must lie in nuclear gene polymorphisms. Identification of the mechanistic underpinnings of the accelerated

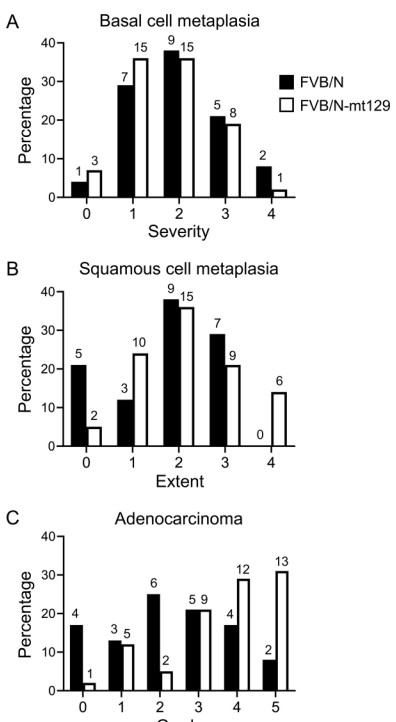

**Fig 3. Percentage of mice in each group having the specified phenotypic abnormalities.** (A) Basal cell metaplasia severity. (B) Squamous cell metaplasia extent. (C) Adenocarcinoma grade. FVB/N, black bars; FVB/N-mt129, white bars. Number of mice is indicated above bars.

**Table 2. Histological characterization of uterine phenotypic abnormalities.**

| Phenotype | Rating | FVB/N (N = 24) | FVB/N-mt129 (N = 42) |
|---|---|---|---|
| | | n (%) | n (%) |
| Basal cell metaplasia severity | 0 | 1 (4) | 3 (7) |
| | 1 | 7 (29) | 15 (36) |
| | 2 | 9 (38) | 15 (36) |
| | 3 | 5 (21) | 8 (19) |
| | 4 | 2 (8) | 1 (2) |
| | 5 | 0 (0) | 0 (0) |
| Squamous cell metaplasia extent | 0 | 5 (21) | 2 (5) |
| | 1 | 3 (13) | 10 (24) |
| | 2 | 9 (38) | 15 (36) |
| | 3 | 7 (29) | 9 (21) |
| | 4 | 0 (0) | 6 (14) |
| | 5 | 0 (0) | 0 (0) |
| Adenocarcinoma grade | 0 | 4 (17) | 1 (2) |
| | 1 | 3 (13) | 5 (12) |
| | 2 | 6 (25) | 2 (5) |
| | 3 | 5 (21) | 9 (21) |
| | 4 | 4 (17) | 12 (29) |
| | 5 | 2 (8) | 13 (31) |

cancer phenotype could be explored further using mouse lines with diverse genetic backgrounds, such as the Collaborative Cross or Diversity Outbred mice [30,31].

Somewhat unexpectedly, the FVB/N-mt129S6 line had a mild but statistically significant increase in severity of DES-induced endometrial cancer. A potential explanation for this finding is incompatibility between the FVB/N nuclear genome and the 129S6 mitochondrial genome, a concept referred to as "cyto-nuclear incompatibility" [32,33]. Cyto-nuclear incompatibility can lead to differences in diverse phenotypes including energetic capacity and cognition, so an impact on cancer development is possible [34,35]. Overall, our findings highlight the utility of the conplastic strain in future studies of endometrial cancer and perhaps other mouse cancer models. In fact, now that the conplastic mouse line exists, it will be possible to shift any existing congenic FVB/N mutant mouse lines onto the conplastic FVB/N-mt129 background in one generation. Thus, the FVB/N-mt129 mice and their pluripotent and readily editable ES cell line will serve as important resources to the scientific community as an isogenic genetic background alternative to C57BL/6 or 129-dervived ES cells. Additionally, the favorable reproductive characteristics maintained in the FVB/N-mt129 conplastic mouse line are an added benefit that can accelerate studies that require larger numbers of mice or for which the genotypes sought are relatively rare despite using optimal breeding strategies.

## Supporting information

**S1 Table. MiniMUGA results from conplastic mice.** After ten generations of backcross, ten conplastic FVB/N-mt129 mice were analyzed by miniMUGA and compared to the FVB/NJ genotype. Variants were grouped based on consistency with FVB/NJ genotype: "Yes" (99.1%; 8865/8830), if all ten mice matched the FVB/N genotype, "Partial" (0.7%, 62/8830), if all mice that had a reported genotype matched FVB/NJ but some mice failed to yield a genotype call for the specific variant, or "No" (18/8830, 0.2%); if some mice had reported variants that were not consistent with FVB/NJ genotype. (XLSX)

## Acknowledgments

The authors would like to thank the Molecular Genomics Core and the Comparative Medicine Branch Animal Research Support Core for their support.

## Author contributions

**Conceptualization:** Artiom Gruzdev, Wendy N. Jefferson, Carmen J. Williams.

**Data curation:** Artiom Gruzdev.

**Formal analysis:** Rani S. Sellers.

**Funding acquisition:** Carmen J. Williams.

**Investigation:** Artiom Gruzdev, Wendy N. Jefferson, Thomas B. Hagler, Gregory J. Scott, Manas K. Ray, Ginger W. Muse, Rani S. Sellers.

**Methodology:** Artiom Gruzdev, Wendy N. Jefferson, Rani S. Sellers.

**Project administration:** Artiom Gruzdev, Carmen J. Williams.

**Supervision:** Carmen J. Williams.

**Visualization:** Rani S. Sellers.

**Writing – original draft:** Artiom Gruzdev, Carmen J. Williams.

**Writing – review & editing:** Wendy N. Jefferson, Thomas B. Hagler, Gregory J. Scott, Manas K. Ray, Ginger W. Muse, Rani S. Sellers.

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
