## [Decision Letter · Decision Letter 0]

28 Oct 2025

Dear Dr. Williams,

Thank you for submitting your manuscript to PLOS ONE. After careful consideration, we feel that it has merit but does not fully meet PLOS ONE’s publication criteria as it currently stands. Therefore, we invite you to submit a revised version of the manuscript that addresses the points raised during the review process.

We look forward to receiving your revised manuscript.

Kind regards,

Benjamin Benzon, Ph.D., M.D.

Academic Editor

PLOS ONE

Journal Requirements:

“This work was supported by the Intramural Research Program of the National Institutes of Health (NIH), National Institute of Environmental Health Sciences, 1ZIAES102405 (CJW) and 1ZICES102425 (Gene Editing and Mouse Model Core).”

“This research was supported by the Intramural Research Program of the National Institutes of 354 Health (NIH), National Institute of Environmental Health Sciences, 1ZIAES102405 (CJW) and 355 1ZICES102425 (Gene Editing and Mouse Model Core). The contributions of the NIH author(s) 356 are considered Works of the United States Government. The findings and conclusions 357 presented in this paper are those of the author(s) and do not necessarily reflect the views of the 358 NIH or the U.S. Department of Health and Human Services. The authors would like to thank the 359 Molecular Genomics Core (1ZICES102546) and the Comparative Medicine Branch Animal 360 Research Support Core (1ZIGES102585) for their support”

“This work was supported by the Intramural Research Program of the National Institutes of Health (NIH), National Institute of Environmental Health Sciences, 1ZIAES102405 (CJW) and 1ZICES102425 (Gene Editing and Mouse Model Core).”

Additional Editor Comments:

The reviewers made some specific comments, please address them in logical and coherent manner.

I am looking forward to your reply and revised manuscript.

Reviewer's Responses to Questions

**Comments to the Author**

1. Is the manuscript technically sound, and do the data support the conclusions?

Reviewer #1: Yes

Reviewer #2: Partly

2. Has the statistical analysis been performed appropriately and rigorously?

Reviewer #1: Yes

Reviewer #2: No

3. Have the authors made all data underlying the findings in their manuscript fully available?

Reviewer #1: Yes

Reviewer #2: Yes

4. Is the manuscript presented in an intelligible fashion and written in standard English?

Reviewer #1: Yes

Reviewer #2: Yes

Reviewer #1: This manuscript tested an interesting hypothesis whether an SNP in the FVB/N strain contributes to the accelerated onset of endometrial cancer in the DES-induced EC model. The authors used an elegant conplastic genetic model to test this hypothesis and found that there was no difference in cancer incidence but surprisingly found higher cancer grade in the conplastic strain. The authors conclude that the increased EC susceptibility lies in nuclear gene polymorphisms but not the SNP in the FVB/N mitochondrial genome. This is an important research addressing the role of strain differences in response to DES-induced EC development. The conplastic mouse strain generated is a very valuable resource to researchers in the field of cancer and developmental biology research. There are some clarifications that the authors can make to improve this manuscript.

1. The authors could explicitly state the conclusion regarding the role of the SNP on EC incidents more clearly in the abstract.

2. It is intriguing that EC tumor grade is higher in the conplastic uteri. A direct interpretation is that the SNP actually protects uteri from EC progression. It would be nice to discuss this surprising result further by giving some possible mechanisms that can lead to this.

3. Is it known whether or how this SNP alters mitochondrial function in the presence of excessive estrogen signaling?

4. Does the DES-induced EC fall into the newly classified EC subtypes? Endometrial cancer. Nat Rev Dis Primers 7, 88 (2021). This information should be important to be included.

Reviewer #2: The presented manuscript is an original article with novel discoveries in the field of endometrial cancer models and germ line competent embryonic stem cells. However, it does require several improvements in order to be suitable for publishing in PLOS one. In short, I believe that the biggest shortcomings of the paper are the presentation and analysis of the Results, as well as Discussion where some oversights have taken place. The statistical analysis should also be explained in detail.

Commens throughout the text by row:

Row 33 – Change “diethylstilbesterol” to “diethylstilbestrol”.

Row 52 – You state: “leading to altered mitochondrial function”

Such as? What are the altered functions and what are they connected to other then ROS production? How does it potentially drive endometrial cancer? I believe this is an important information to elaborate on since the whole paper is based on the fact that the mitochondria is, according to the Abstract, “mildly disfunctional”.

Row 60-62 – You state: “To test this idea, we generated a conplastic strain of FVB/N mice that carried 61 129S6/SvEvTac mitochondria (FVB/N-mt129S6/SvEvTac; referred to henceforth as FVB/N62mt129).”

Why did you use this mouse strain? Any other mouse strain could have been used. Consider writing a sentence or two about why this mouse strain was chosen. (Perhaps due to its potential to generate embryonic stem cells?)

Row 91 – Consider changing “FVB/N-mt 129” to “FVB/N-mt129S6/SvEvTac” due to the naming in Abstract. It should be uniform. Or put “FVB/N-mt129S6/SvEvTac (FVB/N-mt 129)”.

Row 94 – You state: “Females were then backcrossed for 17 generations (N17)”

Why 17?

Row 199 – Please indicate the volume of vehicle/corn oil.

Row 206 – You state: “Mice were humanely euthanized at 9 months of age.”

How did you choose this timepoint? Especially considering that in ROWS 44 and 45 you offer several potential time points.

Row 208 – You state: “FVB/N mice, N=24; FVB/N-mt129 mice, N=42”

How come there is almost twice as much of FVB/N-mt129 mice in the experiment? Please elaborate.

Row 208 – You state: “Two sections 36 μm apart were microscopically evaluated”

Were they both scored? It is not explained in the “Histological analysis” section. Please elaborate.

Row 253 – “Statistical analysis”

This part should be written in much more detail, especially because your sample size is extremely different between groups. You should provide detailed explanation of statistical analysis for each method.

Row 324 – You state: “This difference did not quite reach significance” replace with “This difference was not statistically significant”

Row 297-298 – You state: “FVB/N-mt129 conplastic mouse line readily produced ES cells using blastocyst

298 plating methodology”

Did you try to obtain them from FVB/N mouse line considering that Reference 23 provides an improved protocol? If no, explain why. If yes, please include the Results of the experiment.

Figure 2 – The quality of the images is quite poor. Please increase the quality. Also, the Figure would benefit from better labelling. Maybe consider circling or enlarging the important parts of tissue.

Figure 3 – The results should be presented in a better way than the three graphs presented. Due to the difference in group sizes, the graphs are hard to interpret. Please find a suitable alternative.

Additional comments:

Considering the complexity of the experiment, I believe that the paper would benefit from schematic representation of the experiment timeline.

Do you have any pictures of the animals or uterus (perhaps on a scale paper, or with a ruler) after sacrifice in order to provide morphological appearance of the uterus and associated tissue? The Figure 2 would look more informative.

Other than tumor grade, it would be beneficial if the study included additional tests such as CT scans of the mice to track tumor progress throughout time, or just prior to sacrifice. In that way, a more accurate image of tumor size and spreading could have been obtained. Why weren’t any other characterization methods other than basic histology done?

Maybe the abstract should be adjusted more towards the “embryonic stem cells” part considering that the major portion of the paper’s content is directed more towards obtaining stem cells than on actually comparing the two animal models.

How did you characterise the stem cells? Why is that not explained and provided in the Methods and Results?

Even though there is a substantial amount of stem cells mention, there are no Figures relating to either obtaining or cultivation. Please consider including additional Figure relating to stem cells in the Manuscript.

The authors also didn’t fully exclude the impact of mitochondrial genome since none of the experiments “post model establishment” focused on assessing mitochondrial function and ROS production between the strains. Why weren’t ROS measurements performed?

Considering the unexpected results regarding tumor development in constrained mice, it would be beneficial to discuss this issue in more detail. Please comment on possible causes and provide references if possible.

Consider separating Results from Discussion.

Go once again through the text and uniform words and phrases (e.g. Fig. or Figure, germ line or germline…)

**Do you want your identity to be public for this peer review?** For information about this choice, including consent withdrawal, please see our Privacy Policy

Reviewer #1: **Yes:** Liang Ma

Reviewer #2: No

---

## [Author Response · Author response to Decision Letter 1]

9 Dec 2025

We appreciate the detailed review carried out by the reviewers and feel that the manuscript has been strengthened by their comments. A point-by-point response to the reviewer comments is provided here and the needed text and Figure revisions incorporated into the manuscript. A clean manuscript file and a second manuscript file with changes tracked are provided.

Reviewer #1: This manuscript tested an interesting hypothesis whether an SNP in the FVB/N strain contributes to the accelerated onset of endometrial cancer in the DES-induced EC model. The authors used an elegant conplastic genetic model to test this hypothesis and found that there was no difference in cancer incidence but surprisingly found higher cancer grade in the conplastic strain. The authors conclude that the increased EC susceptibility lies in nuclear gene polymorphisms but not the SNP in the FVB/N mitochondrial genome. This is an important research addressing the role of strain differences in response to DES-induced EC development. The conplastic mouse strain generated is a very valuable resource to researchers in the field of cancer and developmental biology research. There are some clarifications that the authors can make to improve this manuscript.

1. The authors could explicitly state the conclusion regarding the role of the SNP on EC incidents more clearly in the abstract.

Response: This comment highlights an unexpected finding in our study that will be the focus of future work. In terms of germline capable stem cell generation, the idea behind generating a conplastic mouse line is straightforward – shifting from mitochondria found in a non-permissive strain to mitochondria in the classically most-permissive strain. Because ultimately, we plan to use this conplastic mouse line and ES cells to generate complex genetic models in the pure FVB/N nuclear genome background, we needed to confirm that the mitochondrial variant shift did not attenuate DES-induced tumor formation. Once we completed blinded pathological scoring, we found that tumor formation was not attenuated in the conplastic females, but potentially increased. Given that our objective was to confirm DES induced tumor formation was still occurring, and our limited sample size, we are hesitant to overstate any causative relationship between the mitochondrial variants and tumor formation. However, this will be a subject of future studies by our group because the potential to have a more responsive DES-induced cancer model has many research use benefits.

2. It is intriguing that EC tumor grade is higher in the conplastic uteri. A direct interpretation is that the SNP actually protects uteri from EC progression. It would be nice to discuss this surprising result further by giving some possible mechanisms that can lead to this.

Response: Although it is possible that the SNP could protect from EC progression, we think it is more likely that cytoplasmic-nuclear incompatibility contributed to the EC progression rather than the SNP being protective. This concept is now raised in the Results and Discussion section and is relevant to a question from Reviewer 2 as well.

3. Is it known whether or how this SNP alters mitochondrial function in the presence of excessive estrogen signaling?

Response: We could find no published literature reporting whether this SNP alters mitochondrial function under the influence of estrogen.

4. Does the DES-induced EC fall into the newly classified EC subtypes? Endometrial cancer. Nat Rev Dis Primers 7, 88 (2021). This information should be important to be included.

Response: DES-induced uterine adenocarcinoma does not correlate with any of the molecular subgroups described in the Makker et al. reference provided above. It is an endometrioid adenocarcinoma with no known mutations. Molecular characterization of the DES-induced cancer using single cell RNA-seq showed widespread activation of Wnt/β-catenin signaling in epithelial cells and abnormal PI3K/AKT activation in cancer cells. This information has been added to the Introduction.

Reviewer #2: The presented manuscript is an original article with novel discoveries in the field of endometrial cancer models and germ line competent embryonic stem cells. However, it does require several improvements in order to be suitable for publishing in PLOS one. In short, I believe that the biggest shortcomings of the paper are the presentation and analysis of the Results, as well as Discussion where some oversights have taken place. The statistical analysis should also be explained in detail.

Commens throughout the text by row:

Row 33 – Change “diethylstilbesterol” to “diethylstilbestrol”.

Response: Change made in text.

Row 52 – You state: “leading to altered mitochondrial function”

Such as? What are the altered functions and what are they connected to other then ROS production? How does it potentially drive endometrial cancer? I believe this is an important information to elaborate on since the whole paper is based on the fact that the mitochondria is, according to the Abstract, “mildly disfunctional”.

Response: The altered mitochondrial function is documented by an increase in the ATP/ADP ratio despite normal ATP levels and elevated ROS. It is the elevated ROS that could drive endometrial cancer. This point has now been clarified in the Abstract (by removing reference to mitochondrial dysfunction other than ROS) and in the Introduction where the mitochondrial variant phenotype is described.

Row 60-62 – You state: “To test this idea, we generated a conplastic strain of FVB/N mice that carried 61 129S6/SvEvTac mitochondria (FVB/N-mt129S6/SvEvTac; referred to henceforth as FVB/N62mt129).”

Why did you use this mouse strain? Any other mouse strain could have been used. Consider writing a sentence or two about why this mouse strain was chosen. (Perhaps due to its potential to generate embryonic stem cells?)

Response: The 129S6 genetic background was chosen because one of the primary study objectives was to generate germline capable ES cells, and 129S6 was similar to the 129/Ola genetic background, from which the E14 ES cells were originally derived. A short explanation of the rationale for using 129S6 has been added to the Introduction.

Row 91 – Consider changing “FVB/N-mt 129” to “FVB/N-mt129S6/SvEvTac” due to the naming in Abstract. It should be uniform. Or put “FVB/N-mt129S6/SvEvTac (FVB/N-mt 129)”.

Response: Changed in Abstract and Methods as suggested.

Row 94 – You state: “Females were then backcrossed for 17 generations (N17)”

Why 17?

Response: Ultimately, seventeen generations was due to timing and logistics. The gold standard to backcross a nuclear allele from one genetic background to another genetic background takes ten generations, at which point 99.9% of unlinked loci should be of the recipient genetic background. Although we had no intentional nuclear locus selection in our study, we chose roughly the same generational time point to generate our FVB-mt129 ES cells. It took time from first generation of ES clones (N11), to testing their germline contribution competency (N13), to genetically modifying the germline capable clones (N14), to confirming that genetically modified FVB-mt129 ES cells maintain their germline contribution potential (N16). Once we confirmed that we indeed generated gene editable germline capable FVB/N nuclear genome ES cells, we generated a cohort of mice to confirm that DES-induced tumor formation was similar to the parental FVB/N mouse line. During this entire process, we did not want to risk inadvertently developing a recombinant inbred background by in-crossing FVB-mt129 mice, so the primary FVB-mt129 mouse line was maintained by perpetual backcross of conplastic females to isogenic FVB/NJ males. Although this timing could be conceptually interesting to some, we do not believe a full description is worthy of inclusion in the manuscript.

Row 199 – Please indicate the volume of vehicle/corn oil.

Response: Volume of corn oil added to the Methods as suggested.

Row 206 – You state: “Mice were humanely euthanized at 9 months of age.”

How did you choose this timepoint? Especially considering that in ROWS 44 and 45 you offer several potential time points.

Response: We selected 9 months as our timepoint to be able to determine any potential difference between the two lines (increased or decreased). For that reason, we wanted to attain >50% incidence. Given that the incidence at 8 months was 46% in one study and 49% in another study, we decided to wait one additional month to achieve >50%. We have added this explanation in the Methods section.

Row 208 – You state: “FVB/N mice, N=24; FVB/N-mt129 mice, N=42”

How come there is almost twice as much of FVB/N-mt129 mice in the experiment? Please elaborate.

Response: These were naturally bred time-pregnant mice for both groups. Unfortunately, we only had 32 females in the wild type FVB/N group (n=4 litters of 8 female pups) and one of those litters was lost due to flooding prior to weaning and was not included in the experiment (now mentioned in the Methods), resulting in n=3 litters of 8 female pups. We randomly obtained far more females than males in the FVB/N-nt129 group, resulting in n=5 litters of 8-9 female pups. The power calculation to observe a difference with 80% confidence was n=22; this information is added to the Methods in the Statistical Analysis section. Because we had n=24, the power analysis indicated that we had enough to answer the question. In addition, the cancer incidence for the wild type FVB/N mice has been published and the group with lower numbers was the wild type group; the cancer incidence reported in this manuscript is consistent with previous findings.

Row 208 – You state: “Two sections 36 μm apart were microscopically evaluated”

Were they both scored? It is not explained in the “Histological analysis” section. Please elaborate.

Response: Both sections were scored and were typically not different, but in case of differences the more severe score was recorded. This procedure is now stated in the methods.

Row 253 – “Statistical analysis”

This part should be written in much more detail, especially because your sample size is extremely different between groups. You should provide detailed explanation of statistical analysis for each method.

Response: We have added details to the Statistical analysis section as suggested. The Fisher’s exact text is appropriate for this sample size and for different sample numbers between groups.

Row 324 – You state: “This difference did not quite reach significance” replace with “This difference was not statistically significant”

Response: Change made in text as suggested.

Row 297-298 – You state: “FVB/N-mt129 conplastic mouse line readily produced ES cells using blastocyst

298 plating methodology”

Did you try to obtain them from FVB/N mouse line considering that Reference 23 provides an improved protocol? If no, explain why. If yes, please include the Results of the experiment.

Response: We did not attempt to obtain ES cells from the FVB/N mouse line for the following reasons. First, “ES cells”, i.e. non-senescing blastocyst-derived cell lines, can be readily generated using various protocols from mice from nearly all genetic backgrounds, though their having germline competency is more challenging. In fact, at one point Jackson Laboratory sold FVB/NJ mouse ES cells with the disclaimer that germline competency was unknown. Second, in reference 23 (Singh et al., Am J Stem Cells 2012), the B6FVBF1 hybrid ES cells generated had C57BL/6, not FVB, mitochondria. The improved protocol in reference 23 is likely irrelevant for us because C57BL/6-derived mouse ES cells have been generated from nearly all C57BL/6-subtypes using standard mouse ES cell establishment protocols. Third, the FVB genetic background has been extensively used in cancer biology research for over 35 years, and prior to the advent of gene editing directly in single cell embryos with targetable nuclease, gene editing was only possible in mouse ES cells with standard chimera founder generation. However, no groups have ever published the generation of genetically modified mouse lines made directly in pure FVB nuclear genome mouse ES cells. Fourth, although the generation of isogenic FVB/N mouse ES cells is relatively inexpensive, downstream attempts to use them to generate chimera founder mice to test their germline competency would be a substantial cost both in labor and mouse resources (financially and ethically). Fifth, anecdotally in the gene editing world, the FVB genetic background has been unofficially considered non-permissive in generation of germline capable mouse ES cells. Empirically proving that pure FVB mouse ES cells can not readily form germline transmitting founder mice – proving a negative – is practically a Sisyphean task. That said, the reviewer comment is valid, and we note that the lack of parallel pure isogenic FVB/N mouse ES cell germline competency testing is a limitation in our study. We have added a short note about this limitation to the Results & Discussion section.

Figure 2 – The quality of the images is quite poor. Please increase the quality. Also, the Figure would benefit from better labelling. Maybe consider circling or enlarging the important parts of tissue.

Response: The submitted Figure 2 file is a 13.6 Mb .tif that is 6.15 x 8.59 inches and 300 dpi resolution. The figure is of outstanding quality when I look at the original, with no pixelation even with significant zooming in, and I suspect this reviewer was looking at a low-resolution pdf, which would not be used in the final manuscript.

The goal of the low magnification images was to provide a sense of range in extent of disease. Because of the diffuse nature of the cancer, it is not possible to circle all the important parts of the tissue in each panel without obscuring the actual histology, so we used small arrows or arrowheads to highlight the locations instead. We also provided a high magnification set of panels (G-J) that show the various histological patterns observed in this tissue following DES exposure.

We have added a reference for a published paper that provides a full description of the DES-induced endometrial cancer histology in the Methods next to the Figure 2 description.

Figure 3 – The results should be presented in a better way than the three graphs presented. Due to the difference in group sizes, the graphs are hard to interpret. Please find a suitable alternative.

Response: We have now presented the data as a percentage of mice in each category and have included the number of mice above each bar.

Additional comments:

Considering the complexity of the experiment, I believe that the paper would benefit from schematic representation of the experiment timeline.

Response: We do not feel that a schematic of the experimental timeline, which is a standard 5 day neonatal treatment followed by a single time point for analysis, would be useful for most readers.

Do you have any pictures of the animals or uterus (perhaps on a scale paper, or with a ruler) after sacrifice in order to provide morphological appearance of the uterus and associated tissue? The Figure 2 would look more informative.

Response: We do not have gross pictures of the uterus upon collection as the tumors are diffuse and focal in nature and cannot be observed by gross visualizations.

Other than tumor grade, it would be beneficial if the study included additional tests such as CT scans of the mice to track tumor progress throughout time, or just prior to sacrifice. In that way, a more accurate image of tumor size and spreading could have been obtained. Why weren’t any other characterization methods other than basic histology done?

Response: This model of neonatal DES induced uterine cancer has been very well characterized (Suen, et al. 2018 Differentiation Patterns of Uterine Carcinomas and Precursor Lesions Induced by Neonatal Estrogen Exposure in Mice. Toxicologic Pathology 46(5):574–96); this reference is provided in the text. The goal of the experiment was to determine if non-FVB mitochondria on an FVB background impacted the incidence of uterine cancer. We selected a time point that we know from s

---

## [Decision Letter · Decision Letter 1]

8 Jan 2026

Conplastic FVB/N-mt129S6/SvEvTac mice: A new tool for cancer research

PONE-D-25-48629R1

Dear Dr. Williams,

We’re pleased to inform you that your manuscript has been judged scientifically suitable for publication and will be formally accepted for publication once it meets all outstanding technical requirements.

Kind regards,

Benjamin Benzon, Ph.D., M.D.

Academic Editor

PLOS One

Additional Editor Comments (optional):

Reviewers' comments:

Reviewer's Responses to Questions

**Comments to the Author**

Reviewer #1: (No Response)

Reviewer #2: All comments have been addressed

2. Is the manuscript technically sound, and do the data support the conclusions?

Reviewer #1: Yes

Reviewer #2: Yes

3. Has the statistical analysis been performed appropriately and rigorously?

Reviewer #1: Yes

Reviewer #2: Yes

4. Have the authors made all data underlying the findings in their manuscript fully available?

Reviewer #1: Yes

Reviewer #2: Yes

5. Is the manuscript presented in an intelligible fashion and written in standard English?

Reviewer #1: Yes

Reviewer #2: Yes

Reviewer #1: (No Response)

Reviewer #2: (No Response)

**Do you want your identity to be public for this peer review?** For information about this choice, including consent withdrawal, please see our Privacy Policy

Reviewer #1: No

Reviewer #2: No

---

## [Editor Report · Acceptance letter]

PONE-D-25-48629R1

PLOS One

Dear Dr. Williams,

I'm pleased to inform you that your manuscript has been deemed suitable for publication in PLOS One. Congratulations! Your manuscript is now being handed over to our production team.

Kind regards,

on behalf of

Dr. Benjamin Benzon

Academic Editor

PLOS One